# Novel Antiviral Strategies in the Treatment of COVID-19: A Review

**DOI:** 10.3390/microorganisms8091259

**Published:** 2020-08-20

**Authors:** Shiu-Jau Chen, Shao-Cheng Wang, Yuan-Chuan Chen

**Affiliations:** 1Department of Neurosurgery, Mackay Memorial Hospital, Taipei 10449, Taiwan; chenshiujau@gmail.com; 2Department of Medicine, Mackay Medical College, New Taipei City 25245, Taiwan; 3Jianan Psychiatric Center, Ministry of Health and Welfare, Tainan 71742, Taiwan; WShaocheng@gmail.com; 4Department of Mental Health, Johns Hopkins Bloomberg School of Public Health, Baltimore, MD 21205, USA; 5Gloria Operation Center, National Tsing Hua University, Hsinchu City 30013, Taiwan; 6Program in Comparative Biochemistry, University of California, Berkeley, CA 94720, USA

**Keywords:** coronaviruses, COVID-19, SARS–CoV-2, compassionate use, convalescent plasma, CRISPR, mesenchymal stem cell therapy

## Abstract

The coronavirus disease 2019 (COVID-19) pandemic, caused by severe acute respiratory syndrome coronavirus-2 (SARS–CoV-2), is still a global public health problem for humans. It has caused more than 10,000,000 infections and more than 500,000 deaths in the world so far. Many scientists have tried their best to discover safe and effective drugs for the treatment of this disease; however, there are still no approved standard therapeutics or effective antiviral drugs on the market. Many new drugs are being developed, and several traditional drugs that were originally indicated or proposed for other diseases are likely to be effective in treating COVID-19, but their safety and efficacy are controversial, under study, or in clinical trial phases. Fortunately, some novel antiviral strategies, such as convalescent plasma, clustered regularly interspaced short palindromic repeats (CRISPR), and mesenchymal stem cell (MSC) therapy, potentially offer an additional or alternative option or compassionate use for the people suffering from COVID-19, especially for critically ill patients, although their safety and efficacy are also under study. In this review, we explore the applications, possible mechanisms, and efficacy in successful cases using convalescent plasma, CRISPR, and MSC therapy for COVID-19 treatment, respectively. Furthermore, the perspectives and limitations of these novel antiviral strategies are evaluated.

## 1. Introduction

### 1.1. Coronavirus Overview

Coronaviruses are enveloped viruses that are spherical in shape and characterized by crown-like spikes on their surface. This type of virus can be further divided into four subgroups: alpha, beta, gamma, and delta. Coronaviruses, containing many coronavirus strains, are major pathogens in both humans and other animals. Seven strains can cause human diseases, including two alpha coronaviruses (human CoV-229E (HCov-229E) and HCoV-NL63), five beta coronaviruses (HCov-HKU1, HCov-OC43, middle east respiratory syndrome coronavirus (MERS–CoV), severe acute respiratory syndrome coronavirus (SARS–CoV)), and the newly discovered SARS–CoV-2, which is responsible for the coronavirus disease 2019 (COVID-19) pandemic [1,2].

### 1.2. SARS–CoV-2 Molecular Biology

The genetic element of SARS–CoV-2 is a positive sense, single-stranded RNA (ssRNA). The virion has a spirally symmetrical capsid and envelope, and its genome size is about 30 kilobase, which is the largest of all RNA viruses. There is a rod-shaped spike glycoprotein on the envelope, which can be connected to the angiotensin-converting enzyme 2 (ACE2) receptor on the human cell surface [3]. Following the specific binding of the receptor-binding domain of the spike glycoprotein to the ACE2 receptor of the host cell, the entry of the viral single-stranded RNA genome is facilitated [4]. The host cells begin to synthesize replicases and transcriptases to form a replicase–transcriptase complex, including RNA-dependent RNA polymerase, RNA helicase, RNA 5′-triphosphatase, exoribonuclease, N7-methyltransferase, 2′-omethyltransferase, to replicate viral RNA and synthesize viral structural proteins. The newly synthesized RNA and structural proteins assemble together to form mature virions. Finally, the virions are sent to the cell surface in the form of vesicles and then released from the original host cells via exocytosis to infect new host cells (Figure 1) [5,6].

### 1.3. COVID-19 Outbreak, Transmission, and Clinical Severity

At the beginning of the 21st century, several viral diseases emerged and raised the attention of both medical and public health researchers, including SARS from 2002 to 2004, H1N1 influenza in 2009, and MERS from 2012 to 2015. Recently, COVID-19 in Wuhan city, in China’s Hubei province, was first reported as an epidemic to the World Health Organization (WHO) on 31 December 2019. WHO then announced COVID-19 as a public health emergency of international concern (PHEIC) on 12 February 2020.

The infection of SARS–CoV-2 can be brought about by droplet transmission or direct contact with secretions from coughing or sneezing of COVID-19 patients and may be followed by clinical symptoms and/or illness. According to the intensity of respiratory dysfunction, COVID-19 has a different clinical spectrum, such as uncomplicated (mild) illness, moderate pneumonia, severe pneumonia, acute respiratory distress syndrome (ARDS), and even sepsis, septic shock, and multiple organ dysfunction syndromes. The main symptoms include fever, dry cough, fatigue, expectoration, shortness of breath, muscle pain or joint pain, sore throat, headache, chills, nausea or vomiting, stuffy nose, diarrhea, coughing up blood, and conjunctival hyperemia. Additionally, abnormal smell and/or taste have been reported. Most people infected with SARS–CoV-2 will experience mild to moderate respiratory illness, and some of them will recover without requiring special treatment. However, serious respiratory diseases may follow, including pneumonia in some cases. Older people and those with underlying medical problems such as cardiovascular disease, diabetes, chronic respiratory disease, and cancer are a high-risk population and more likely to develop a serious illness, which may even lead to death in some cases [7].

The disease control, diagnosis, treatment, and prevention of COVID-19 are very difficult due to the following characteristics of SARS–CoV-2 [8]. (1) Cross-species transmission: It can be transmitted between different species of animals and between people and animals, or among people [8]. (2) Diagnostic variability: Although there may be a negative diagnosis after repeated tests, it may become positive unexpectedly [8]. (3) Multiple transmission routes: It is mainly transmitted through droplets or direct contact, but some studies have even found that it can be transmitted by air [9,10,11,12], feces [13,14,15], and asymptomatic carriers [16,17,18]. Moreover, its basic reproductive number (R0) has been determined to range between 2–3^2^, and the spreading region is probably wide enough to trigger a pandemic [19]. (4) Genome mutation possibility: The mutation of the viral RNA genome is possible [12,19]; hence, the research and development of vaccines and drugs are great challenges. Even if the discovery is successful and products are approved for marketing, the effective period of the vaccine products and approved drugs may be limited due to mutation and resistance.

## 2. Supportive Therapeutic and Traditional Drugs

The high incidence of respiratory failure and high mortality rates result in a heavy burden on medical services, as well as the urgent need of potential medications for COVID-19. However, no specific antiviral treatment or vaccine for COVID-19 is currently available; that is, there is no evidence from randomized controlled trials to support specific drug treatment for COVID-19. At present, the definite therapeutic strategies to deal with the infection are only supportive therapies, which have limited efficacy. For example, symptomatic treatment and oxygen therapy can be used to prevent acute respiratory failure [20,21]. If oxygen therapy is not helpful, noninvasive and invasive mechanical ventilation (IMV), high flow nasal oxygen therapy, and even extracorporeal membrane oxygenation (ECMO) should be considered for the patients with rapidly progressive respiratory failure [22,23,24]. Regarding traditional ARDS treatment, Gattinoni et al. suggested that COVID-19-induced ARDS is not a “typical” ARDS [25], and IMV should be postulated earlier to avoid excessive intrathoracic negative pressures and lung injury. The use of systemic corticosteroids for ARDS or viral pneumonia is not recommended; however, for patients with rapid progression of respiratory failure, methylprednisolone could be considered for use appropriately [26,27]. If there is any possible accompanying bacterial pneumonia, antibacterial drugs are recommended.

Despite this, some traditional drugs originally indicated or proposed for other diseases may be effective for the treatment of COVID-19. Potential therapeutic options of COVID-19, such as antimalarials, antivirals against viruses, or cytokines against symptoms, are under study or clinical trials, but their safety and efficacy are still unclear or controversial (Table 1) [28].

## 3. Novel Antiviral Strategies

A number of publications have addressed possible therapeutics of COVID-19, but very few reports have confirmed the safety and effectiveness of these therapeutics for the treatment of this disease. No standard treatment for COVID-19 has been approved, and most of the traditional drugs for inhibiting SARS–CoV-2 are still under evaluation. Consequently, some novel antiviral strategies can probably be helpful for the treatment of COVID-19 and reduction of the death rate of this disease, especially for patients who are at the most dangerous phase of this disease.

### 3.1. Convalescent Plasma

Convalescent plasma is referred to as immune plasma collected from patients who have recovered from infectious diseases. The transfusion of convalescent plasma may contain passive antibodies and provide immediate passive immunity to susceptible individuals for the short-term. Meanwhile, the current pandemic of COVID-19 and indefinite efficacy of traditional drugs have called attention to convalescent plasma as a novel antiviral strategy to treat this disease. The pathogenesis of COVID-19 is probably affected by direct neutralization, control of an overactive immune system (e.g., cytokine storm, Th1/Th17 ratio, complement activation) by anti-inflammatory and immunoregulatory activity, and immunomodulation of a hypercoagulable state, which are implemented by the components of convalescent plasma [52]. Several examples have shown that convalescent plasma has been successfully used as postexposure prophylaxis and/or treatment for COVID-19 [53,54,55,56,57]. Although the data are limited, the data suggest clinical benefits in the aspects of radiological resolution, reduction in viral loads, and improved survival in the hope of reducing morbidity and mortality [58,59].

Shen et al. used convalescent plasma transfusion obtained from five patients who had recovered from COVID-19 to treat five critically ill patients who had COVID-19 and ARDS [53]. The convalescent plasma contained SARS–CoV-2-specific antibodies, where the IgG-binding titer was greater than 1:1000 and the neutralization titer was greater than 40. Although these patients had received antiviral treatment and mechanical ventilation, they still presented severe pneumonia with fast progression and continuously high viral loads [53]. Following the receipt of convalescent plasma, three out of five patients were discharged, with the remaining two patients in stable condition at 37 days after transfusion. In this case series study of five critically ill patients with COVID-19 and ARDS, the administration of convalescent plasma containing neutralizing antibodies significantly improved their clinical status [53]. However, the limited sample size and study design preclude a definitive conclusion about the potential efficacy, and these observations are required to be evaluated in clinical trials.

Duan et al. transfused one dose of 200 mL of convalescent plasma with the neutralizing antibody, derived from recently recovered donors, into 10 patients with COVID-19 to explore the possibility of convalescent plasma transfusion to rescue severe patients [54]. After convalescent plasma transfusion, the level of neutralizing antibody remained at a high level (1:640) in nine cases (one case was unavailable). The clinical symptoms and paraclinical criteria were significantly improved within three days. Compared with pretransfusion, several clinical data were improved: lymphocyte counts were increased, C-reactive protein was decreased, varying degrees of absorption of lung lesions were shown in radiological examinations, and the viral load was not detected in seven patients who previously had viremia [54]. Additionally, there were no serious adverse effects that emerged during this period. These results revealed that convalescent plasma was able to serve as a promising choice for severe COVID-19 patients [54]. However, larger, well-controlled, and randomized trials are required for further evaluation of the optimal dose, time point, and clinical benefits.

Zhang et al. used anti-SARS–CoV-2 antibodies, including IgM and IgG measured by enzyme-linked immunosorbent assays (ELISAs), in convalescent plasma from six donors who had recovered from COVID-19 to test if convalescent plasma can be utilized for the treatment of severe COVID-19 patients [55]. The recipient was a 64-year-old female. Eleven days after convalescent plasma transfusion, they found that she did not require mechanical ventilation and could be transferred to a general ward. The results revealed that the convalescent plasma may facilitate insight into the SARS–CoV-2 infection and established donor screening protocol for COVID-19 [55]. However, the efficacy of convalescent plasma is still indefinite because the favorable outcome was achieved in only one patient (sample size is small), and the clinical outcome may have been confounded by other concomitant treatments.

Ye et al. did a descriptive study to evaluate the efficacy of convalescent plasma therapy in patients with COVID-19 [56]. Six patients were enrolled and received convalescent plasma transfusions. The efficacy of convalescent plasma was determined by the alleviation of symptoms, changes in radiologic abnormalities, and laboratory tests. They found that the transfusion of convalescent plasma resulted in a resolution of ground-glass opacities (GGOs) and consolidation in five patients, and viruses were removed in two patients. The serologic analysis indicated anti-SARS–CoV-2 antibody titers immediately increased in two patients. Moreover, no severe side effects were observed. The results showed that convalescent plasma was effective, specific, and significant for the removal of SARS–CoV-2 [56]. However, the study is neither universal nor representative due to the very small sample size.

Ahn et al., in a case report, used convalescent plasma to treat two patients with COVID-19 who presented severe pneumonia with ARDS [57]. In addition to systemic corticosteroid administration, both patients showed a favorable outcome without any severe adverse effects after convalescent plasma transfusion. The cases suggest that convalescent plasma might be an additional or alternate option to treat patients. Moreover, it is possible to simultaneously reduce viral loads by convalescent plasma and an excessive inflammatory response by systemic corticosteroids [57]. However, more scientific evidence and clinical trials are required to prove the efficacy and safety of convalescent plasma. Additionally, the given number of antibodies to each patient should be standardized, and other treatments that may influence the correlation between convalescent plasma and antibody should be investigated.

### 3.2. CRISPR

The clustered regularly interspaced short palindromic repeats (CRISPR)/CRISPR associated protein 9 (Cas9) system contains two components: Cas9, an endonuclease, and a single-guide RNA (sgRNA) that guides Cas9 to a specific location in the genome. Cas9 will unwind the DNA duplex and cut both strands as a target sequence is recognized by sgRNA. By the cooperation of a specifically designed sgRNA and Cas9, the genome can be cleaved at most locations with only the availability of a protospacer adjacent motif (PAM) sequence (NGG) that is 3 nucleotides upstream from the target site [60,61,62]. Being a gene-targeting technology, CRISPR has also been successfully used as an antiviral approach for the elimination of a variety of viruses [60]. Therefore, this suggests that the CRISPR can potentially act as a novel antiviral strategy for the treatment of COVID-19.

Abbott et al. designed and screened CRISPR-associated RNAs (crRNAs) to target conserved viral regions and identify functional crRNAs targeting SARS–CoV-2 in cell culture [63]. PAC-MAN (prophylactic antiviral CRISPR in human cells) is a CRISPR/Cas13-based strategy for RNA-guided viral RNA inhibition and degradation. In human lung epithelial cells, the RNAs from SARS–CoV-2 sequences and live influenza A virus (IAV) can be effectively degraded using PAC-MAN [63]. Their bioinformatic analysis revealed that a group of six crRNAs could target more than 90% of all coronaviruses. The results demonstrated that PAC-MAN decreased about 90% of viral loads and may be useful for targeting a variety of viruses, and could become a powerful approach in blocking viral replication and gene expression. With the development of a safe and effective delivery system in the respiratory tract, PAC-MAN has the potential to become a crucial strategy for the inhibition of pancoronaviruses [63]. However, this study was only performed in cell culture, and in vivo studies and further clinical trials are needed.

### 3.3. Mesenchymal Stem Cell Therapy

Cell-based approaches, primarily using mesenchymal stem cells (MSCs), have demonstrated to provide a possible safe and effective novel antiviral strategy for patients with ARDS [64,65]. Immunomodulatory, regenerative, and anti-inflammatory properties of MSC treatment have been proposed as a suitable therapeutic approach, and several clinical trials have begun for COVID-19 treatment. A set of various cell sources, doses, administration strategies, and target patient populations have been applied in MSC therapy [65]. After MSC transplantation, the immunomodulatory effect of MSCs could protect alveolar epithelial cells, reclaim the pulmonary microenvironment, prevent pulmonary fibrosis, and cure lung dysfunction because of the significant cell accumulation in the lungs [64,65]. Currently, no MSC-based therapy has been approved for the prevention and/or treatment of COVID-19, but related clinical trials are being continuously conducted now.

Leng et al. intravenously injected MSC into seven enrolled patients to investigate whether MSC transplantation is effective in treating COVID-19 [66]. They evaluated the patients’ clinical outcomes, inflammatory changes, immune function levels, and adverse effects for 14 days. After MSC transplantation, all patients’ functional outcomes were significantly improved, without any observed adverse effects. The peripheral lymphocytes were increased, and the C-reactive protein was decreased. Moreover, the overactivated cytokine-secreting immune cells CXCR3 + CD4 + T-cells, CXCR3 + CD8 + T-cells, and CXCR3 + NK cells disappeared within 3–6 days, and regulatory dendritic cells (DCs) were dramatically increased [66]. Compared with the placebo group, the level of TNF-α was decreased and IL-10 was increased in the MSC treatment group. The clinical outcome of patients was significantly improved, which may have been due to the regulation of inflammatory response and the promotion of tissue repair and regeneration induced by MSCs [66]. The results showed that MSC transplantation was both safe and effective, especially for critically ill COVID-19 patients.

Atluri et al. described the pathogenesis of coronaviruses, the urgent need for various solutions to COVID-19, and the clinical evidence regarding its treatment using stem cells [67]. The role of expanded umbilical cord mesenchymal stem cells (UC–MSCs) in treating COVID-19 is being studied. The patient’s own immune system is critical to curing COVID-19. The immune system triggers the production of many inflammatory factors and causes a severe cytokine storm if it is overactivated in the process of destroying viruses. The cytokine storm may impair organs and result in edema, air exchange dysfunction, ARDS, acute cardiac injury, secondary infection, and may even lead to death. The limited evidence related to UC–MSCs in managing COVID-19 suggests that compassionate therapy should only be used in critically ill patients to reduce morbidity and mortality [67].

## 4. Perspectives and Limitations

At this stage, the most prospective application of novel antiviral strategies is compassionate use (expanded access) for COVID-19, like Remdesivir [29]. Compassionate use is a possible way for patients with immediate life-threatening conditions or serious diseases to receive investigational medical products, including chemicals, biopharmaceuticals, or medical devices for treatment, free from the limitation of undergoing clinical trials. These investigational medical products have not yet been approved or cleared, and their safety and effectiveness for their specific use have not been confirmed. The investigational medical product may or may not be effective in the treatment of the condition, and their use may cause unexpected and serious side effects. However, under the supervision of an institutional review board (IRB) and with the informed consent of patients, medical products being developed can be made available to patients who have no comparable or satisfactory therapy options. Compassionate use may be appropriate when the following requirements apply (Table 2) [68,69].

### 4.1. Convalescent Plasma

Convalescent plasma transfusion is considered to be one of the most ancient and well-known therapeutic methods, but it is seldom used for disease treatment due to safety concerns such as rejection, allergy, contamination with microbes, and mixture with nonspecific proteins from donors [70,71,72,73,74,75]. Moreover, its efficacy in treating diseases is challenging and controversial [70,71,72,73,74,75]. Recently, with the emergency and severity of the COVID-19 pandemic, convalescent plasma has been tried in some critically ill patients to rescue their lives [58,59,76,77]. Some successful cases (described in Section 3.1) may promote convalescent plasma as a prospective strategy for the treatment of COVID-19 [53,54,55,56,57]. Blood centers have been established to be firm infrastructure for collecting and constructing stocks of convalescent plasma to meet the globally growing demand [58].

However, there are still some limitations for the use of convalescent plasma to treat COVID-19, as follows [53,54,55,56,57,58,59,76,77,78]: (1) The sample size of the research is too small to be representative; all patients who have successfully used convalescent plasma for therapy are only special and/or specific cases, not universal cases. (2) Scientific evidence is not enough because large-scale clinical trials that can represent the target patient populations (e.g., age, race, gender, and comorbid disease) are lacking or still ongoing. (3) The number of antibodies that the patients received was not standardized. (4) The patient usually received convalescent plasma and other treatments (e.g., antiviral agents, steroids) concomitantly. This may have affected the association between convalescent plasma and the antibody, resulting in confusing therapeutic results. (5) The results for patient prognosis and surveillance post-treatment are still unavailable or insufficient; they must be monitored for a long period of time to evaluate the efficacy and side effects carefully. (6) Data from rigorously controlled clinical trials are too few, namely, the range of indications (e.g., prevention vs. treatment), as well as an overview of benefits. Risk and regulation adaptability need to be considered deliberately.

### 4.2. CRISPR

Along with the development of a safe and effective gene targeting approach (e.g., CRISPR) for respiratory tract delivery, PAC-MAN has the potential to become a perspective strategy for pancoronavirus inhibition. The pancoronavirus protection using CRISPR-Cas13d will offer an alternative and complementary strategy over traditional antiviral drugs or vaccines.

However, the biggest barrier to clinically utilizing PAC-MAN is the development of in-vivo delivery tools that are safe, efficient, and effective. Moreover, there are several technical limitations for using PAC-MAN against SARS–CoV-2 in clinical applications, as follows [63]: (1) PAC-MAN functions in a cell-autonomous way, suggesting only cells that express Cas13d and crRNAs are directly protected against the targeted viruses. (2) PAC-MAN needs full expression in a certain percentage of cells, and the range should be determined experimentally prior to use in patients. (3) The competency to adjust Cas13d levels and its associated crRNAs in cells may be critical to efficiently inhibit viruses, in that viral sequence cleavage efficiency is sensitive to crRNA expression. (4) Viral genomes may be resistant to inhibition due to the RNA secondary structure or protective protein coats; thus, high-throughput screening of crRNAs may be needed for the identification of highly effective crRNAs targeting live viruses. (5) The risk of mutation or tumorigenesis induced by off-target effects must be avoided and completely removed before undergoing clinical trials [79,80,81]. (6) The selection of a suitable delivery tool that is safe, specific, and efficient in delivering CRISPR to target cells in patients is not easy [82,83]. The patients would have more unique benefits than traditional antiviral drugs or vaccines when using the PAC-MAN strategy to treat COVID-19 if these aforementioned obstacles are overcome.

### 4.3. MSC Therapy Strategy

The process of developing new therapeutic strategies and promoting MSC-based therapy for clinical application has important and practical implications for the treatment of COVID-19. COVID-19 is so serious and human lives are valuable; hence, the effectiveness of therapeutic preparation using MSC therapy is becoming a hot topic. The MSC secretome can provide a novel therapeutic approach for the treatment of COVID-19 because of its extensive pharmacological effects, such as anti-inflammation, immunomodulation, regeneration, proangiogenicity, and antifibrosis [84]. Based on this evidence, the MSC secretome administered by intravenous injection or inhalation is likely to become a promising antiviral strategy for the treatment of COVID-19, particularly for patients in critically severe conditions [66,84]. To maximize potential therapeutic use, it is critical to understand the related preclinical studies and postulated mechanisms of MSCs in respiratory lung injuries induced by viruses. Though the clinical application of MSC therapy to treat COVID-19 has not been approved, some promising reports have been recently presented [64,65]. Therefore, stem cell therapy, especially MSCs, may be one of the most prospective therapeutics or a combination of treatments for patients with COVID-19.

However, MSC therapy has only been proven to be safe and effective on a specific, special, and limited basis. The case series study, where seven patients were enrolled, has ensured the feasibility and safety of MSC therapy, but the clinical efficacy needs to be confirmed in random controlled trials, including more cases, to have significant clinical statistics [66]. Moreover, these patients’ prognosis should be monitored for the long term to achieve definite evidence of clinical efficacy [66]. The administration and Coronavirus Task Force might hope to use expanded UC–MSCs as an evolutionary therapeutic strategy in managing COVID-19, with pronged approaches as follows [67]: (1) All agencies should minimize the regulatory burden so that critically ill patients will have access without considering their financial source or support. (2) Appropriate safeguards instituted should refrain from negative results from immoral individuals. (3) When patients are in an urgent situation, and with their proper informed consent, the approach can be tried in critically ill patients who have no response to traditional drugs.

## 5. Conclusions

The COVID-19 pandemic has caused serious global economic, political, public health, and even military problems for the past year, and the consequences are currently persisting in many countries. During this period, almost every country has taken measures for disease prevention, such as quarantines, isolation, disinfection, border blockades, traffic cutoffs, social distancing, homestay, and restriction of passenger entry, to prevent a COVID-19 epidemic outbreak. There are currently no approved effective drugs or licensed vaccine products for COVID-19. Under the condition that traditional drugs cannot assure their safety and efficacy for COVID-19 treatment, novel antiviral strategies, including convalescent plasma, CRISPR, and cell therapy, may be able to provide an additional or alternative option or compassionate use for the treatment of COVID-19, particular for critically ill patients.

## Figures and Tables

**Figure 1 microorganisms-08-01259-f001:**
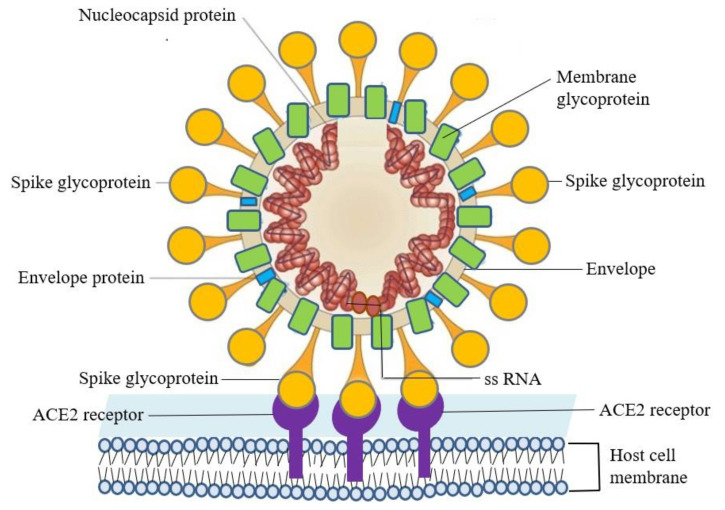
SARS–CoV-2 infects host cells through the specific binding of spike glycoprotein on the viral envelope and the angiotensin-converting enzyme 2 (ACE2) receptor on the host cell’s surface.

**Table 1 microorganisms-08-01259-t001:** Potentially useful traditional drugs (originally indicated or proposed for other diseases) for COVID-19.

Type	Drug	Component	Mode of Action	Original Indication or Proposed Usage
Anti-virus	Remdesivir [29,30,31,32,33]	Adenosine nucleotide analogs	Viral application inhibitor	A broad-spectrum antiviral against Ebola viruses
	Hydroxy-chloroquine, Chloroquine [34,35,36,37]	4-aminoquinoline	Change the pH of endosomes and prevents viral entry, transport and postentry events	Malaria prophylaxis, autoimmune disease (e.g., rheumatic diseases)
	Lopinavir/Ritonavir [38,39]	C_37_H_48_N_4_O_5_/C_37_H_48_N_6_O_5_S_2_	Protease inhibitor	Acquired immunodeficiency syndrome (AIDS)
	Favipiravir [40,41]	Pyrazine carboxamide derivative	RNA polymerase inhibitors	Influenza
	Ivermectin [42]	Mixture of 22, 23-dihydroavermectin B1a and 22, 23-dihydroavermectin B1b	Nuclear import of host and viral proteins inhibitor	Parasite infestation including head lice, scabies, river blindness (onchocerciasis), strongyloidiasis, trichuriasis, ascariasis, lymphatic filariasis
Anti-symptom	Tocilizumab [43,44,45], Sarilumab [46], Eculizumab [47]	Monoclonal antibody	Interleukin-6 (IL-6) inhibitor, to block cytokine storm	Rheumatoid arthritis
	Bevacizumab [48,49]	Monoclonal antibody	Vascular endothelial growth factor A (VEGF-A) inhibitor	Angiogenesis of cancer cells
	Ruxolitinib [50,51]	(3*R*)-3-cyclopentyl-3-[4-(7*H*-pyrrolo [2,3-d]pyrimidin-4-yl)pyrazol-1-yl] propanenitrile	Janus kinase 1/2 inhibitor	Myelofibrosis

**Table 2 microorganisms-08-01259-t002:** The requirements for compassionate use recommended by the United States Food and Drug Administration (US FDA) and the European Medicines Agency (EMA).

Institution	Requirement
US FDA	The patient has a serious disease or condition, or whose life is immediately threatened by these disorders.
	No comparable or satisfactory alternative therapy is available to diagnose, monitor, or treat the disease or condition.
	It is impossible for patients to enroll in a clinical trial.
	Potential benefit justifies the potential risks of treatment for patients.
	The investigational medical product will not interfere with clinical trials that could support the development or marketing approval for the indication of this product.
EMA	Patients have serious diseases with no satisfactorily authorized therapies and cannot enter clinical trials.
	The programs are only put in place if the medicine is expected to help patients with life-threatening, long-lasting, or seriously debilitating illnesses.
	The medicine must be undergoing clinical trials or have entered the marketing-authorization application process.
	Early studies of the medicine will generally have been completed, though its safety profile and dosage guidelines may not be fully established.

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
