# Peer review of "Novel Antiviral Strategies in the Treatment of COVID-19: A Review"

_microorganisms, 2020, doi:10.3390/microorganisms8091259_

Round 1
Reviewer 1 Report
While there is merit and interest in the content of the review paper, more efforts should be taken to ensure that information are paraphrased even if references are cited.
Under Coronavirus Overview (page 2):
Lines 44 – 45: There is a rod-shaped spike glycoprotein on the envelope, which can be connected to the angiotensin converting enzyme 2 (ACE2) receptor on the human cell surface.
Comment: References 3 and 4 were published before COVID-19 and did not mention that ACE2 is the receptor for SARS-CoV-2.
Lines 46 : The sentence should be corrected to “ Following the binding of receptor binding domain of the spike glycoprotein to the ACE2 receptor of host cell, entry of viral single stranded RNA genome is facilitated.”
Under COVID-19 Outbreak, Transmission and Clinical Severity (pages 2 – 3)
Lines 60 -61: “the SARS..the MERS”, “the” should be removed.
Lines 62 : reported as an epidemic to WHO
Line 65 : Infection of SARS-CoV2 can be brought about by droplet ..or direct contact..
Line 77 : cancer are at risk population and are more likely
Lines 82 : Sentence structure should be changed.
Lines 83 : It is mainly transmitted by..
Line 84: some studies have even found that it can be transmitted by air, feces and asymptomatic carriers
Comment: Reference needed.
Line 85: More elaboration needed
Lines 87: ( R & D) not needed
Line 86: The mutation rate of viral RNA genome is high
Comment: Reference 6 suggests the possibility of mutation but did not state that mutation rate is high. Please check
Under Supportive Therapeutic and Traditional Drugs (pages 3 – 4)
Lines 91 : High incidence of respiratoy failure and morality result in heavy burden on medical services
Lines 94: At present, the definite therapeutic strategies to deal with the infection are only supportive therapies which have limited efficacy.
Lines 96-100 For example, symptomatic treatment and oxygen therapy can be used to prevent the acute respiratory failure. If oxygen therapy is not helpful, non-invasive (NIV), invasive mechanical ventilation (IMV), high flow nasal oxygen therapy (HFNO), and even extracorporeal membrane oxygenation (ECMO) should be considered for the patients with rapidly progressive respiratory failure.
Comment: References needed. Abbreviations are not needed if this is only used once.
Lines 100-101 : Sentence structure should be changed.
Lines 102 – 105: The use of systemic corticosteroids for ARDS or viral pneumonia were not recommended; however, for patients with rapid progression of respiratory failure, methylprednisolone could be considered to use appropriately. If there is any possible accompanying bacterial pneumonia, antibacterial drugs are recommended.
Comment: References needed.
Lines 107-108 : Sentence structure should be changed.
Table 1
Comment: Please check reference 15 to ensure that contents are paraphrased in article. Reference 18 may not be relevant in Table 1 as it suggests that hydroxychloroquine has side-effects and its use should be considered carefully. References 24 – 26 are for Tocilizumab only.
Lines 113 : A number of publications ..
Lines 116 : “are still under evaluation” instead of “still being discovered or tried”.
Lines 116 -118 : “ Consequently….” . Long sentence, please split into 2.
Under Convalescent Plasma (pages 5 – 6)
Lines 120 : who have recovered
Lines 122 : provide immediate.. sentence structure should be changed.
Lines 124 – 127: The possible mechanisms of convalescent plasma and influence in COVID-19 pathogenesis include direct neutralization, control of an overactive immune system (e.g., cytokine storm, Th1/Th17 ratio, complement activation) and immunomodulation of a hypercoagulable state [31].
Comment: Needs to be rephrased as phrasing is identical to reference 31.Also please elaborate on how active immune system is controlled.
Lines 127 : several examples have shown… references are needed
Lines 128 : Although instead of though, data is limited
Lines 129-130 : references are needed
Lines 132 – 133: had severe pneumonia with rapid progression and continuously high viral loads despite antiviral treatment and mechanical ventilation [32]. The SARS-CoV-2-specific antibodies (IgG binding titer was greater than 1:1000 and a neutralization titer was greater than 40) were obtained from 5 patients who had recovered from COVID-19.
Comment: Needs to be rephrased or quoted as phrasing is identical to reference 32.
Lines 135 : Following the receipt of convalescent plasma, 3 out of 5 patients were discharged with the remaining 2 in stable condition. Initial phrasing similar to ref 32, please check.
Line 137 : Not sure what is preliminary uncontrolled case series
Lines 144 – 151: After convalescent plasma transfusion, the level of neutralizing antibody increased rapidly up to 1:640 in 5 cases, while that of the other 4 cases maintained at a high level (1:640). The clinical symptoms and paraclinical criteria were significantly improved within 3 day. Compared with pretransfusion, several parameters were improved including increased lymphocyte counts and decreased C-reactive protein. Radiological examinations showed varying degrees of absorption of lung lesions within 7 day. The viral load was undetectable after transfusion in 7 patients who had previous viremia. No severe adverse effects were observed during this period.
Comment: Needs to be rephrased or quoted as phrasing is identical/very similar to reference 33.
Lines 152 – 153: However, the optimal dose and time point. as well as the clinical benefit needs further investigation in larger well-controlled and randomized trials.
Comment: Needs to be rephrased or quoted as phrasing is identical to reference 33.
Lines 157 – 158: treated with convalescent plasma did not require mechanical ventilation 11 days after transfusion. and was then transferred to a general ward. The serological findings in convalescent plasma may facilitate insight into the SARS-CoV-2 infection and establish donor screening protocol for COVID-19 [34]. However, the efficacy of convalescent plasma is still inconclusive because the sample size is too small and other concomitant treatments might confound the result.
Comment: Needs to be rephrased or quoted as phrasing is identical/very similar to reference 34.
Lines 165 – 168: The efficacy of this intervention was determined by the alleviation of symptoms, changes in radiologic abnormalities and laboratory tests. No obvious adverse effect observed during the treatment. Transfusion of convalescent plasma resulted in a resolution of ground glass opacities (GGOs) and consolidation in 5 patients.
Comment: Needs to be rephrased or quoted as phrasing is identical/very similar to reference 35.
Lines 170 – 172: The results indicated that convalescent plasma was effective and specific for COVID-19. This intervention was significant for the elimination of SARS-CoV-2 and promising to be a state-of-art therapy [35].
Comment: Needs to be rephrased or quoted as phrasing is identical/very similar to reference 35.
Lines 177 – 178: The results showed that convalescent plasma has been expected to increase survival rates as in the case with emerging SARS-CoV-2 infections.
Comment: Reference 36 suggested that convalescent plasma has been expected to increase survival rates due to promising results from other viral infections e.g. MERS and this formed the motivation behind the study instead of it being the outcome.
Lines 179 – 183: Moreover, it is possible to reduce excessive inflammatory response by corticosteroids and viral loads by convalescent plasma simultaneously. when used with systemic corticosteroids [36]. However, more well-designed studies and clinical trials are needed to demonstrate the efficacy and safety of convalescent plasma transfusion in COVID-19 patients.
Comment: Needs to be rephrased or quoted as phrasing is very similar to reference 36.
Under CRISPR (page 6)
Lines 185 – 189: In the clustered regularly interspaced short palindromic repeats (CRISPR)/CRISPR-associated protein (Cas) system, CRISPR is used to build RNA-guided genes drives to target a specific DNA sequence. By the Cas proteins and a specifically designed single-guiding RNA (sgRNA), the genome can be cleaved at most locations with only the limitation of a protospacer adjacent motif (PAM) sequence (NGG) is available in the target site [37 – 39].
Comment: Needs to be rephrased or quoted as phrasing is identical/very similar to reference 39.
Lines 189 – 190: Being a gene targeting technology, CRISPR has also been successfully used as an antiviral agent for the elimination of a variety of viruses [30].
Comment: Reference 30 did not describe CRISPR.
Lines 196 – 197: PAC-MAN can effectively degrade RNA from SARS-CoV-2 sequences and live influenza A virus (IAV) in human lung epithelial cells.
Comment: Needs to be rephrased or quoted as phrasing is identical/very similar to reference 40.
Under Cell Therapy (pages 6 – 7)
Lines 210 – 211: A range of different cell sources, doses, dosing strategies and targeted patient populations have been applied.
Comment: Needs to be rephrased or quoted as phrasing is identical to reference 42.
Lines 211 – 215: After the MSC transplantation, a significant population of cells accumulates in the lung, which their immunomodulatory effect could protect alveolar epithelial cells, reclaim the pulmonary microenvironment, prevent pulmonary fibrosis and cure lung dysfunction [41. 42]. Currently, there are no approved MSC-based therapy for the prevention and/or treatment of COVID- 19, but clinical trials are ongoing.
Comment: Needs to be rephrased or quoted as phrasing is identical/very similar to reference 41.
Lines 217 – 229: The clinical outcomes. inflammatory changes. immune function levels and adverse effects of patients were evaluated for 14 days. MSCs could significantly improve the functional outcomes of all patients without any observed adverse effects. After MSC transplantation, the peripheral lymphocytes were increased and the C-reactive protein was decreased. Moreover, the over-activated cytokine-secreting immune cells CXCR3+CD4+ T cells, CXCR3+CD8+ T cells and CXCR3+ NK cells were disappeared in 3-6 days. Additionally, a group of regulatory dendritic cells (DC) population were dramatically increased. Also, the level of TNF-α was decreased, while IL-10 was increased in MSC treatment group compared with the placebo group. Furthermore, MSCs were ACE2- and TMPRSS2- in the gene expression profile to demonstrate to be free from SARS-CoV-2 infections. The results showed that the reasons why MSC transplantation significantly improved the outcome of COVID-19 patients may be the regulation of inflammatory response and promotion of tissue repair and regeneration [43]. MSC transplantation was safe and effective. especially for the critically ill COVID-19 patients.
Comment: Needs to be rephrased or quoted as phrasing is identical/very similar to reference 43.
Lines 230 – 231: Atluri et al. described the urgent need for various solutions. pathogenesis of coronavirus and the clinical evidence for treatment of COVID-19 with stem cells [44].
Comment: Needs to be rephrased or quoted as phrasing is identical to reference 44.
Lines 232 – 236: The cure of COVID-19 must be rely on the patients' own immune system. If the immune system is over-activated in an attempt to kill viruses, it will trigger the production of many inflammatory factors, leading to severe cytokine storm. The cytokine storm may damage organs followed by the edema, air exchange dysfunction, ARDS, acute cardiac injury and secondary infection, which may result in death.
Comment: Needs to be rephrased or quoted as phrasing is identical/very similar to reference 44.
Under Perspectives and Limitations (pages 7 – 8)
Lines 241 – 244: Compassionate use is a potential pathway for patients with immediately life-threatening conditions or serious diseases to gain access to investigational medical products including chemicals, biopharmaceuticals or medical devices for treatment outside of clinical trials.
Comment: Needs to be rephrased or quoted as phrasing is very similar to reference 45.
Under Convalescent Plasma (page 8)
Lines 252 – 253: it is seldom used for disease treatment due to the safety concerns such as rejection, allergy, contamination with microbes, mixture with nonspecific proteins from donors
Comment: Reference needed
Lines 257 – 259: The blood centers have robust infrastructure for collecting and constructing inventories of convalescent plasma to meet the globally growing demand.
Comment: Needs to be rephrased or quoted as phrasing is very similar to reference 47.
Lines 263 – 268: (2) Scientific evidence is not enough because of the lack of large scale clinical trials able to represent the target patient populations (e.g., age, race, gender and comorbid disease, etc.). (3) The number of antibodies administered to each patient was not standardized. (4) Convalescent plasma usually proceeds with other treatments, such as antiviral agents and steroids, which can affect the correlation between convalescent plasma and antibody. confounding the results.
Comment: Needs to be rephrased or quoted as phrasing is very similar to reference 36.
Under CRISPR (page 8)
Lines 280 – 288: (1) PAC-MAN functions in a cell autonomous manner. suggesting only cells that express Cas13d and crRNAs are directly protected against the targeted viruses. (2) For PAC-MAN to be effective in patients, it needs to sufficiently expressed in a certain percentage of cells, the range need to be determined experimentally. (3) Because SARS-CoV-2 sequence cleavage efficiency is sensitive to crRNA expression, the ability to adjust Cas13d level and its cognate crRNAs in the cell may be critical to achieve efficient viral inhibition. (4) Viral genomes may be less susceptible to inhibition due to the secondary structure of the RNA genome or coating with protective proteins; thus, high-throughput screening of crRNAs may help identify highly effective crRNAs targeting live SARS-CoV-2.
Comment: Needs to be rephrased or quoted as phrasing is identical/very similar to reference 40.
Lines 288 – 290: The risk of mutation or tumorigenesis induced by off target effects must be avoided and completely removed before undergoing clinical trials.
Comment: Reference needed.
Lines If these barriers are surmountable, there are unique benefits over traditional antiviral drugs or vaccines. using the PAC-MAN strategy therapeutically.
Comment: Needs to be rephrased or quoted as phrasing is identical/very similar to reference 40.
Under Cell Therapy (page 9)
Lines 296 – 298: MSC secretome can provide a novel therapeutic approach in treating COVID-19 due to the broad pharmacological effects including anti-inflammatory. Immunomodulatory, regenerative, pro-angiogenic and anti-fibrotic properties.
Comment: Needs to be rephrased or quoted as phrasing is identical to reference 51.
Lines 299 – 300: MSC-secretome formulated as a freeze-dried powder and given by intravenous injection or inhalation
Comment: Needs to be rephrased or quoted as phrasing is identical to reference 51.
Lines 308 – 314: The administration and Coronavirus Task Force might hope to approach the potential of expanded UC-MSCs as an evolutionary therapeutic strategy in managing COVID-19 with pronged approaches as follows [44]: (1) Minimize regulatory burden by all agencies so that critically ill patients will have access regardless of their financial support. (2) Institute appropriate safeguards should avoid negative consequences from unscrupulous actors. (3) With proper informed consent from patients or when patients are in need and subject to accumulate data in that cohort, the procedure will be allowed to be tried in critically ill patients who have no responses to traditional drugs.
Comment: Needs to be rephrased or quoted as phrasing is identical to reference 44.
Lines 317 – 321: (1) necessary resources for safe administration of cell therapies; (2) determinants of cell therapy utilization; (3) selection among patients with B cell non-Hodgkin lymphomas and B cell acute lymphoblastic leukemia; (4) supportive measures during cell therapy administration; (5) application and prioritization of tocilizumab; and (6) collaborative care with referring physicians.
Comment: Needs to be rephrased or quoted as phrasing is identical to reference 52.
Lines 322 – 325: Because these recommendations are changeable, these questions and recommendations should not serve as reasons to defer CART cell therapy for COVID-19 patients who are truly in need of a potentially curative therapy [52].
Comment: Reference 52 is providing recommendations on providing cell therapies to pediatric relapsed/refractory B-cell acute lymphoblastic leukemia and adult relapsed/refractory diffuse large B cell lymphoma patients safely during COVID-19 and acknowledged that although these recommendations can change as the pandemic progresses, COVID-19 should not be used as a reason to delay this treatment. While CAR T cell therapy is currently being explored for treatment of COVID-19, it was not suggested in reference 52.
Author Response
Responses to the Comment
We thank for the reviewer’ comment and revise our manuscript accordingly in tracked form. In addition, we have this manuscript edited extensively.
Reviewer 1:
While there is merit and interest in the content of the review paper, more efforts should be taken to ensure that information are paraphrased even if references are cited.
Under Coronavirus Overview (page 2):
Lines 44 – 45: There is a rod-shaped spike glycoprotein on the envelope, which can be connected to the angiotensin converting enzyme 2 (ACE2) receptor on the human cell surface.
Comment: References 3 and 4 were published before COVID-19 and did not mention that ACE2 is the receptor for SARS-CoV-2.
Ans: We have added a new reference (reference 3) to mention that ACE2 is the receptor for SARS-CoV-2. (P.2, Line 46)
Lines 46: The sentence should be corrected to “Following the binding of receptor binding domain of the spike glycoprotein to the ACE2 receptor of host cell, entry of viral single stranded RNA genome is facilitated.”
Ans: We have changed the sentence into “Following the specific binding of receptor binding domain of the spike glycoprotein to the ACE2 receptor of host cell, entry of viral single-stranded RNA genome is facilitated.” and added a new reference (reference 4). (P.2, Line 47-48)
Under COVID-19 Outbreak, Transmission and Clinical Severity (pages 2 – 3)
Lines 60 -61: “the SARS. the MERS”, “the” should be removed.
Ans: We have removed “the” before these two words. (P.2, Line 63-64)
Lines 62: reported as an epidemic to WHO
Ans: We have added “as” in this sentence. (P.2, Line 65)
Line 65: Infection of SARS-CoV2 can be brought about by droplet or direct contact.
Ans: We have revised the sentence into “The infection of SARS-CoV-2 can be brought about by droplet transmission or direct contacting with secretions from coughing or sneezing of COVID-19 patients, and may be followed by clinical symptoms and/or illness.” (P.2, Line 68-70)
Line 77: cancer are at risk population and are more likely
Ans: We have revised the sentence into “……and cancer are at risk population and are more likely…….” (P.2, Line 80)
Lines 82: Sentence structure should be changed.
Ans: We have revised these sentences into “(2) Diagnostic variability: Although it shows negative in diagnosis after repeated tests, it may diagnose to be positive unexpectedly.” (P.2, Line 85-86)
Lines 83: It is mainly transmitted by.
Ans: We have revised the sentence into “Multiple transmission routes: It is mainly transmitted by…….”. (P.2, Line 87)
Line 84: some studies have even found that it can be transmitted by air, feces and asymptomatic carriers. Comment: Reference needed.
Ans: We have added ten new references for COVID-19 transmission by air, feces and asymptomatic carrier (Reference 9-18) ((P.2, Line 88-89)
Line 85: More elaboration needed
Ans: We have revised the sentences into “Moreover, its basic reproductive number (R0) has been determined to be ranged 2-32 and the spreading region is probably wide to trigger a pandemic.” (P.2, Line 89-91)
Lines 87: (R & D) not needed
Ans: We have deleted “R&D”. (P.2, Line 90)
Line 86: The mutation rate of viral RNA genome is high
Comment: Reference 6 suggests the possibility of mutation but did not state that mutation rate is high. Please check
Ans: We have revised this sentence into “(4) Genome mutation possibility: The mutation of viral RNA genome is possible” (P.2, Line 91-92)
Under Supportive Therapeutic and Traditional Drugs (pages 3 – 4)
Lines 91: High incidence of respiratory failure and morality result in heavy burden on medical services.
Ans: We have revised the sentences as suggestion. (P.3, Line 97-98)
Lines 94: At present, the definite therapeutic strategies to deal with the infection are only supportive therapies which have limited efficacy.
Ans: We have revised the sentences as suggestion. (P.3, Line 101-102)
Lines 96-100 For example, symptomatic treatment and oxygen therapy can be used to prevent the acute respiratory failure. If oxygen therapy is not helpful, non-invasive (NIV), invasive mechanical ventilation (IMV), high flow nasal oxygen therapy (HFNO), and even extracorporeal membrane oxygenation (ECMO) should be considered for the patients with rapidly progressive respiratory failure.
Comment: References needed. Abbreviations are not needed if this is only used once.
Ans: We have added five new references (Reference 20-24) and removed the abbreviations if they are only used once. (P.3, Line 102-106)
Lines 100-101: Sentence structure should be changed.
Ans: We have revised the sentence into “Gattinoni et al. suggested that COVID-19-induced ARDS is not a "Typical" ARDS, and IMV should be considered to postulate earlier to avoid excessive intrathoracic negative pressures and lung injury.” (P.3, Line 107-109)
Lines 102 – 105: The use of systemic corticosteroids for ARDS or viral pneumonia were not recommended; however, for patients with rapid progression of respiratory failure, methylprednisolone could be considered to use appropriately. If there is any possible accompanying bacterial pneumonia, antibacterial drugs are recommended.
Comment: References needed.
Ans: We have added two new reference (Reference 26-27) (P.3, Line 109-112).
Lines 107-108: Sentence structure should be changed.
Ans: We have revised the sentence into “Potentially therapeutic options of COVID-19 such as antimalarials, antivirals against viruses or cytokines against symptoms are under study or clinical trials, ……” (P.3, Line 115-117)
Table 1
Comment: Please check reference 15 to ensure that contents are paraphrased in article. Reference 18 may not be relevant in Table 1 as it suggests that hydroxychloroquine has side-effects and its use should be considered carefully. References 24 – 26 are for Tocilizumab only.
Ans: In Table 1, we have replaced the original references 15 and 18 with two new references 34 and 37, respectively. Additionally, we also have added two new references for Sarilumab ( Reference 46) and Eculizumab ( Reference 47), respectively.
Lines 113: A number of publications.
Ans: We have revised it as suggestion. (P. 4, Line 121)
Lines 116: “are still under evaluation” instead of “still being discovered or tried”.
Ans: We have revised it as suggestion. (P. 4, Line 124)
Lines 116 -118: “Consequently….”. Long sentence, please split into 2.
Ans: We have split this sentence into “Consequently, some novel antiviral strategies probably be helpful for the treatment of COVID-19 and the reduction of its death rate, especially for patients who are at the most dangerous phase of this disease.” (P.5, Line 124-127)
Under Convalescent Plasma (pages 5 – 6)
Lines 120: who have recovered
Ans: We have revised it as suggestion. (P. 5, Line 129)
Lines 122: provide immediate. sentence structure should be changed.
Ans: We have revised it as suggestion. (P. 5, Line 131)
Lines 124 – 127: The possible mechanisms of convalescent plasma and influence in COVID-19 pathogenesis include direct neutralization, control of an overactive immune system (e.g., cytokine storm, Th1/Th17 ratio, complement activation) and immunomodulation of a hypercoagulable state [31].
Comment: Needs to be rephrased as phrasing is identical to reference 31. Also please elaborate on how active immune system is controlled.
Ans: We have rephrased it as “The pathogenesis of COVID-19 is probably affected by the direct neutralization, control of an overactive immune system (e.g., cytokine storm, Th1/Th17 ratio, complement activation) by anti-inflammatory and immunoregulatory activity, and immunomodulation of a hypercoagulable state, which are implemented by the components of convalescent plasma.” (P.5, Line 136-140)
Lines 127: several examples have shown… references are needed
Ans: We have added the successful cases described in this manuscript as references (Reference 53-57). (P.5, Line 140-142)
Lines 128: Although instead of though, data is limited
Ans: We have revised it as suggestion. (P. 5, Line 142)
Lines 129-130: references are needed
Ans: We have added two new reference (Reference 58, 59). (P.5, Line 144)
Lines 132 – 133: had severe pneumonia with rapid progression and continuously high viral loads despite antiviral treatment and mechanical ventilation [32]. The SARS-CoV-2-specific antibodies (IgG binding titer was greater than 1:1000 and a neutralization titer was greater than 40) were obtained from 5 patients who had recovered from COVID-19.
Comment: Needs to be rephrased or quoted as phrasing is identical to reference 32.
Ans: We have rephrased it and added a quote (Reference 53) as described in P.5, Line 145-154.
Lines 135: Following the receipt of convalescent plasma, 3 out of 5 patients were discharged with the remaining 2 in stable condition. Initial phrasing similar to ref 32, please check.
Ans: We have revised the sentences as suggestion. (P.5, Line 154-157)
Line 137: Not sure what is preliminary uncontrolled case series
Ans: We have revised it into “In this case series study, ……….” (P.5, Line 157-159)
Lines 144 – 151: After convalescent plasma transfusion, the level of neutralizing antibody increased rapidly up to 1:640 in 5 cases, while that of the other 4 cases maintained at a high level (1:640). The clinical symptoms and paraclinical criteria were significantly improved within 3 day. Compared with pretransfusion, several parameters were improved including increased lymphocyte counts and decreased C-reactive protein. Radiological examinations showed varying degrees of absorption of lung lesions within 7 day. The viral load was undetectable after transfusion in 7 patients who had previous viremia. No severe adverse effects were observed during this period.
Comment: Needs to be rephrased or quoted as phrasing is identical/very similar to reference 33.
Ans: We have rephrased it added a quote (Reference 54) as described in P.5, Line 167-P.6, Line 173.
Lines 152 – 153: However, the optimal dose and time point. as well as the clinical benefit needs further investigation in larger well-controlled and randomized trials.
Comment: Needs to be rephrased or quoted as phrasing is identical to reference 33.
Ans: We have rephrased it as described in P.6, Line 175-177.
Lines 157 – 158: treated with convalescent plasma did not require mechanical ventilation 11 days after transfusion. and was then transferred to a general ward. The serological findings in convalescent plasma may facilitate insight into the SARS-CoV-2 infection and establish donor screening protocol for COVID-19 [34]. However, the efficacy of convalescent plasma is still inconclusive because the sample size is too small and other concomitant treatments might confound the result.
Comment: Needs to be rephrased or quoted as phrasing is identical/very similar to reference 34.
Ans: We have rephrased it as described in P.6, Line 181-189.
Lines 165 – 168: The efficacy of this intervention was determined by the alleviation of symptoms, changes in radiologic abnormalities and laboratory tests. No obvious adverse effect observed during the treatment. Transfusion of convalescent plasma resulted in a resolution of ground glass opacities (GGOs) and consolidation in 5 patients.
Comment: Needs to be rephrased or quoted as phrasing is identical/very similar to reference 35.
Ans: We have rephrased it as described in P.6, Line 192-198.
Lines 170 – 172: The results indicated that convalescent plasma was effective and specific for COVID-19. This intervention was significant for the elimination of SARS-CoV-2 and promising to be a state-of-art therapy [35].
Comment: Needs to be rephrased or quoted as phrasing is identical/very similar to reference 35.
Ans: We have rephrased it as described in P.6, Line 198-201.
Lines 177 – 178: The results showed that convalescent plasma has been expected to increase survival rates as in the case with emerging SARS-CoV-2 infections.
Comment: Reference 36 suggested that convalescent plasma has been expected to increase survival rates due to promising results from other viral infections e.g. MERS and this formed the motivation behind the study instead of it being the outcome.
Ans: We have removed the sentence “The results showed that convalescent plasma has been expected to increase survival rates as in the case with emerging SARS-CoV-2 infections.” (P.6, Line 206-207)
Lines 179 – 183: Moreover, it is possible to reduce excessive inflammatory response by corticosteroids and viral loads by convalescent plasma simultaneously. when used with systemic corticosteroids [36]. However, more well-designed studies and clinical trials are needed to demonstrate the efficacy and safety of convalescent plasma transfusion in COVID-19 patients.
Comment: Needs to be rephrased or quoted as phrasing is very similar to reference 36.
Ans: We have rephrased it as described in P.6, Line 208-215.
Under CRISPR (page 6)
Lines 185 – 189: In the clustered regularly interspaced short palindromic repeats (CRISPR)/CRISPR-associated protein (Cas) system, CRISPR is used to build RNA-guided genes drives to target a specific DNA sequence. By the Cas proteins and a specifically designed single-guiding RNA (sgRNA), the genome can be cleaved at most locations with only the limitation of a protospacer adjacent motif (PAM) sequence (NGG) is available in the target site [37 – 39].
Comment: Needs to be rephrased or quoted as phrasing is identical/very similar to reference 39.
Ans: We have rephrased the sentences as “The CRISPR/Cas9 system contains two components: Cas9, an endonuclease, and a single-guide RNA (sgRNA) which guides Cas9 to a specific location in the genome. Cas9 will unwind the DNA duplex and cut both strands as a target sequence is recognized by sgRNA. By the cooperation of a specifically designed sgRNA and Cas9, the genome can be cleaved at most locations with only the availability of a protospacer adjacent motif (PAM) sequence (NGG), which is 3 nucleotides upstream from the target site.” (P.6, Line 221-P.7, 226)
Lines 189 – 190: Being a gene targeting technology, CRISPR has also been successfully used as an antiviral agent for the elimination of a variety of viruses [30].
Comment: Reference 30 did not describe CRISPR.
Ans: We have changed the original reference number 30 to a new number 60. (P.7, Line 228)
Lines 196 – 197: PAC-MAN can effectively degrade RNA from SARS-CoV-2 sequences and live influenza A virus (IAV) in human lung epithelial cells.
Comment: Needs to be rephrased or quoted as phrasing is identical/very similar to reference 40.
Ans: We have rephrased it and added a quote (Reference 63) as described in P.7, Line 233-235.
Under Cell Therapy (pages 6 – 7)
Lines 210 – 211: A range of different cell sources, doses, dosing strategies and targeted patient populations have been applied.
Comment: Needs to be rephrased or quoted as phrasing is identical to reference 42.
Ans: We have rephrased it added a quote (Reference 42) as described in P.7, Line 248-249.
Lines 211 – 215: After the MSC transplantation, a significant population of cells accumulates in the lung, which their immunomodulatory effect could protect alveolar epithelial cells, reclaim the pulmonary microenvironment, prevent pulmonary fibrosis and cure lung dysfunction [41. 42]. Currently, there are no approved MSC-based therapy for the prevention and/or treatment of COVID- 19, but clinical trials are ongoing.
Comment: Needs to be rephrased or quoted as phrasing is identical/very similar to reference 41.
Ans: We have rephrased it as described in P.7, Line 249-255.
Lines 217 – 229: The clinical outcomes, inflammatory changes, immune function levels and adverse effects of patients were evaluated for 14 days. MSCs could significantly improve the functional outcomes of all patients without any observed adverse effects. After MSC transplantation, the peripheral lymphocytes were increased and the C-reactive protein was decreased. Moreover, the over-activated cytokine-secreting immune cells CXCR3+CD4+ T cells, CXCR3+CD8+ T cells and CXCR3+ NK cells were disappeared in 3-6 days. Additionally, a group of regulatory dendritic cells (DC) population were dramatically increased. Also, the level of TNF-α was decreased, while IL-10 was increased in MSC treatment group compared with the placebo group. Furthermore, MSCs were ACE2- and TMPRSS2- in the gene expression profile to demonstrate to be free from SARS-CoV-2 infections. The results showed that the reasons why MSC transplantation significantly improved the outcome of COVID-19 patients may be the regulation of inflammatory response and promotion of tissue repair and regeneration [43]. MSC transplantation was safe and effective. especially for the critically ill COVID-19 patients.
Comment: Needs to be rephrased or quoted as phrasing is identical/very similar to reference 43.
Ans: We have rephrased it added a quote (Reference 66) as described in P.7, Line 257-272.
Lines 230 – 231: Atluri et al. described the urgent need for various solutions, pathogenesis of coronavirus and the clinical evidence for treatment of COVID-19 with stem cells [44].
Comment: Needs to be rephrased or quoted as phrasing is identical to reference 44.
Ans: We have rephrased it as described in P.8, Line 273-275.
Lines 232 – 236: The cure of COVID-19 must be relied on the patients' own immune system. If the immune system is over-activated in an attempt to kill viruses, it will trigger the production of many inflammatory factors, leading to severe cytokine storm. The cytokine storm may damage organs followed by the edema, air exchange dysfunction, ARDS, acute cardiac injury and secondary infection, which may result in death.
Comment: Needs to be rephrased or quoted as phrasing is identical/very similar to reference 44.
Ans: We have rephrased it as described in P.8, Line 276-282.
Under Perspectives and Limitations (pages 7 – 8)
Lines 241 – 244: Compassionate use is a potential pathway for patients with immediately life-threatening conditions or serious diseases to gain access to investigational medical products including chemicals, biopharmaceuticals or medical devices for treatment outside of clinical trials.
Comment: Needs to be rephrased or quoted as phrasing is very similar to reference 45.
Ans: We have rephrased it as described in P.8, Line 287-290.
Under Convalescent Plasma (page 8)
Lines 252 – 253: it is seldom used for disease treatment due to the safety concerns such as rejection, allergy, contamination with microbes, mixture with nonspecific proteins from donors
Comment: Reference needed
Ans: We have added 6 new references (Reference 70-75) to support the statement. (P.8, Line 282)
Lines 257 – 259: The blood centers have robust infrastructure for collecting and constructing inventories of convalescent plasma to meet the globally growing demand.
Comment: Needs to be rephrased or quoted as phrasing is very similar to reference 47.
Ans: We have rephrased it and added a reference (Reference 58) as described in P.9, Line 308-309.
Lines 263 – 268: (2) Scientific evidence is not enough because of the lack of large-scale clinical trials able to represent the target patient populations (e.g., age, race, gender and comorbid disease, etc.). (3) The number of antibodies administered to each patient was not standardized. (4) Convalescent plasma usually proceeds with other treatments, such as antiviral agents and steroids, which can affect the correlation between convalescent plasma and antibody. confounding the results.
Comment: Needs to be rephrased or quoted as phrasing is very similar to reference 36.
Ans: We have rephrased it as described in P.9, Line 313-319
Under CRISPR (page 8)
Lines 280 – 288: (1) PAC-MAN functions in a cell autonomous manner. suggesting only cells that express Cas13d and crRNAs are directly protected against the targeted viruses. (2) For PAC-MAN to be effective in patients, it needs to sufficiently expressed in a certain percentage of cells, the range need to be determined experimentally. (3) Because SARS-CoV-2 sequence cleavage efficiency is sensitive to crRNA expression, the ability to adjust Cas13d level and its cognate crRNAs in the cell may be critical to achieve efficient viral inhibition. (4) Viral genomes may be less susceptible to inhibition due to the secondary structure of the RNA genome or coating with protective proteins; thus, high-throughput screening of crRNAs may help identify highly effective crRNAs targeting live SARS-CoV-2.
Comment: Needs to be rephrased or quoted as phrasing is identical/very similar to reference 40.
Ans: We have rephrased it as described in P.9, Line 332-343.
Lines 288 – 290: The risk of mutation or tumorigenesis induced by off target effects must be avoided and completely removed before undergoing clinical trials.
Comment: Reference needed.
Ans: We have added three references (Reference 79-81) and one sentence including two new references (Reference 82, 83) to support the statement. (P.9, Line 343-346)
Lines 290-291: If these barriers are surmountable, there are unique benefits over traditional antiviral drugs or vaccines. using the PAC-MAN strategy therapeutically.
Comment: Needs to be rephrased or quoted as phrasing is identical/very similar to reference 40.
Ans: We have rephrased it as “The patients would have more unique benefits than traditional antiviral drugs or vaccines using the PAC-MAN strategy to treat COVID-19, if these obstacles had been overcome.” (P.9, Line 347-349)
Under Cell Therapy (page 9)
Lines 296 – 298: MSC secretome can provide a novel therapeutic approach in treating COVID-19 due to the broad pharmacological effects including anti-inflammatory. Immunomodulatory, regenerative, pro-angiogenic and anti-fibrotic properties.
Comment: Needs to be rephrased or quoted as phrasing is identical to reference 51.
Ans: We have rephrased it added a quote (Reference 84) as described in P.10, Line 354-357.
Lines 299 – 300: MSC-secretome formulated as a freeze-dried powder and given by intravenous injection or inhalation
Comment: Needs to be rephrased or quoted as phrasing is identical to reference 51.
Ans: We have rephrased it as described in P.10, Line 358-361.
Lines 308 – 314: The administration and Coronavirus Task Force might hope to approach the potential of expanded UC-MSCs as an evolutionary therapeutic strategy in managing COVID-19 with pronged approaches as follows [44]: (1) Minimize regulatory burden by all agencies so that critically ill patients will have access regardless of their financial support. (2) Institute appropriate safeguards should avoid negative consequences from unscrupulous actors. (3) With proper informed consent from patients or when patients are in need and subject to accumulate data in that cohort, the procedure will be allowed to be tried in critically ill patients who have no responses to traditional drugs.
Comment: Needs to be rephrased or quoted as phrasing is identical to reference 44.
Ans: We have rephrased it as described in P.10, Line 371-380.
Lines 317 – 321: (1) necessary resources for safe administration of cell therapies; (2) determinants of cell therapy utilization; (3) selection among patients with B cell non-Hodgkin lymphomas and B cell acute lymphoblastic leukemia; (4) supportive measures during cell therapy administration; (5) application and prioritization of tocilizumab; and (6) collaborative care with referring physicians.
Comment: Needs to be rephrased or quoted as phrasing is identical to reference 52.
Ans: We decide to remove this paragraph about CART cells and only focus on reviewing mesenchymal stem cell (MSC) therapy.
Lines 322 – 325: Because these recommendations are changeable, these questions and recommendations should not serve as reasons to defer CART cell therapy for COVID-19 patients who are truly in need of a potentially curative therapy [52].
Comment: Reference 52 is providing recommendations on providing cell therapies to pediatric relapsed/refractory B-cell acute lymphoblastic leukemia and adult relapsed/refractory diffuse large B cell lymphoma patients safely during COVID-19 and acknowledged that although these recommendations can change as the pandemic progresses, COVID-19 should not be used as a reason to delay this treatment. While CAR T cell therapy is currently being explored for treatment of COVID-19, it was not suggested in reference 52.
Ans: We decide to remove this paragraph about CART cells and only focus on reviewing mesenchymal stem cell (MSC) therapy.
Reviewer 2:
The authors present a short review of new (novel) antiviral strategies in treating COVID-19.
The review is brief and to the point with the description of novel antiviral therapeutics which include: 1) convalescent plasma, 2) CRISPR, and 3) cell therapy, primarily the utilization of mesenchymal stem cells (MSCs). After a brief discussion of these potential approaches a short presentation of perspectives and limitations regarding each approach is presented followed by a short conclusion section.
The paper is rather slim on data, discussion and conclusions! Granted the field is expanding however this paper would provide a short introduction to a more thorough treatment of “potential new COVID-19 therapies”.
The paper has to be edited for English, specific examples:
Line 113. A plenty of publications.
Ans: We have revised it into “A number of publications”. (P. 5, Line 121)
Line 120. from patients who are recovered.
Ans: We have revised it into “who have recovered”. (P. 5, Line 129)
Line 207. have demonstrated to provide a possible.
Ans: We have revised it into “…. provide a possible safe and effective…….” (P. 7, Line 245)
Line 222. NK cells were disappeared in 3-6 days.
Ans: We have revised it into “…. NK cells were disappeared within 3-6 days….” (P. 7, Line 263)
Line 225 & 226. rewrite
Ans: We have decided to remove this sentence “Furthermore, MSCs were ACE2- and TMPRSS2- in the gene expression profile to demonstrate to be free from SARS-CoV-2 infections.” (P. 7, Line 266-267)
Line 293 - 295. rewrite
Ans: We have rephrased this sentence as “The process of developing new therapeutic strategies and promoting MSC-based therapy to clinical application has important and practical implications for the treatment of COVID-19.” (P. 9, Line 351-353)
Line 304. promising reports have presented recently.
Ans: We have revised it into “…. promising reports have been presented recently….” (P. 10, Line 362-363)

Reviewer 2 Report
The authors present a short review of new (novel) antiviral strategies in treating COVID-19.
The review is brief and to the point with the description of novel antiviral therapeutics which include: 1) convalescent plasma, 2) CRISPR, and 3) cell therapy, primarily the utilization of mesenchymal stem cells (MSCs). After a brief discussion of these potential approaches a short presentation of perspectives and limitations regarding each approach is presented followed by a short conclusion section.
The paper is rather slim on data, discussion and conclusions!
Granted the field is expanding however this paper would provide a short introduction to a more thorough treatment of “potential new COVID-19 therapies”.
The paper has to be edited for English, specific examples:
Line 113.. A plenty of publications..
Line 120.. from patients who are recovered…
Line 207.. have demonstrated to provide a possible..
Line 222.. NK cells were disappeared in 3-6 days..
Line 225 & 226.. rewrite
Line 293 - 295.. rewrite
Line 304.. promising reports have presented recently..
Author Response
Responses to the Comment
We thank for the reviewer’ comment and revise our manuscript accordingly in the tracked form. In addition, we have this manuscript edited extensively.
Reviewer 2:
The authors present a short review of new (novel) antiviral strategies in treating COVID-19.
The review is brief and to the point with the description of novel antiviral therapeutics which include: 1) convalescent plasma, 2) CRISPR, and 3) cell therapy, primarily the utilization of mesenchymal stem cells (MSCs). After a brief discussion of these potential approaches a short presentation of perspectives and limitations regarding each approach is presented followed by a short conclusion section.
The paper is rather slim on data, discussion and conclusions! Granted the field is expanding however this paper would provide a short introduction to a more thorough treatment of “potential new COVID-19 therapies”.
The paper has to be edited for English, specific examples:
Line 113. A plenty of publications.
Ans: We have revised it into “A number of publications”. (P. 5, Line 121)
Line 120. from patients who are recovered.
Ans: We have revised it into “who have recovered”. (P. 5, Line 129)
Line 207. have demonstrated to provide a possible.
Ans: We have revised it into “…. provide a possible safe and effective…….” (P. 7, Line 245)
Line 222. NK cells were disappeared in 3-6 days.
Ans: We have revised it into “…. NK cells were disappeared within 3-6 days….” (P. 7, Line 263)
Line 225 & 226. rewrite
Ans: We have decided to remove this sentence “Furthermore, MSCs were ACE2- and TMPRSS2- in the gene expression profile to demonstrate to be free from SARS-CoV-2 infections.” (P. 7, Line 266-267)
Line 293 - 295. rewrite
Ans: We have rephrased this sentence as “The process of developing new therapeutic strategies and promoting MSC-based therapy to clinical application has important and practical implications for the treatment of COVID-19.” (P. 9, Line 351-353)
Line 304. promising reports have presented recently.
Ans: We have revised it into “…. promising reports have been presented recently….” (P. 10, Line 362-363)

Round 2
Reviewer 2 Report
The authors have responded to the previous review with numerous changes and in general the paper is appropriate for acceptance.
I have noted some specific edits that need addressing, see below.
Line 125—strategies probably may/can(?)…..
Line 152--- had ever had ?? (never)..
Line 188----outcome may be confound (??) by...
Line 255---- conducting (conducted)
Line 263-----disappeared change (disappearing)
Line 333----against not (against )
Line 336----needs to fully expressed change to (…full expression in…)
Line 339--- inhibiting change to (inhibit)
Line 359----these change to (this)
Author Response
Comment
We are grateful for the reviewer’s comment and have revised the manuscript accordingly using the tracked form.
The authors have responded to the previous review with numerous changes and in general the paper is appropriate for acceptance.
I have noted some specific edits that need addressing, see below.
Line 125—strategies probably may/can(?)…..
Ans: We have revised it into “…strategies probably can be …….” (Line 121)
Line 152--- had ever had ?? (never)..
Ans: No, not never. But we have revised it into “…….had ever received…..” (Line 141)
Line 188----outcome may be confound (??) by...
Ans: We have revised it into “…outcome may be confounded by …….” (Line 169)
Line 255---- conducting (conducted)
Ans: We have revised it into “…conducted…….” (Line 222)
Line 263-----disappeared change (disappearing)
Ans: We have revised it into “…disapearing…….” (Line 230)
Line 333----against not (against )
Ans: We have revised it into “…against…….” (Line 293)
Line 336----needs to fully expressed change to (…full expression in…)
Ans: We have revised it into “…full expression in…….” (Line 295-296)
Line 339--- inhibiting change to (inhibit)
Ans: We have revised it into “…inhibit…….” (Line 298)
Line 359----these change to (this)
Ans: We have revised it into “…this….” (Line 325)